

# On local scouring downstream small water structures

Marta Kiraga[1] and Zbigniew Popek[2]

[1] Department of Hydrotechnics, Technology and Management, Institute of Civil Engineering, Warsaw University of Life Sciences WULS-SGGW, Warsaw, Poland
[2] Department of Water Engineering and Applied Geology, Institute of Environmental Engineering, Warsaw University of Life Sciences WULS-SGGW, Warsaw, Poland

## ABSTRACT

**Background**. In order to regulate water flow, hydraulic structures such as weirs or checks, frequently equipped with gates, are used. Water can flow below or over the gate or, simultaneously, over and below the gate. Both diversifications of hydraulic gradient, being an effect of damming up a river by the structure and shear stresses at the bed, which exceeds the critical shear stress value, invoke the local scouring downstream the structure. This phenomenon has been studied in laboratory and field conditions for many years, however Researchers do not agree on the parameters that affect the size of the local scour and the intensity of its formation. There are no universal methods for estimating its magnitude However, solutions are sought in the form of calculation formulas typical for the method of flow through the structure, taking into account the parameters that characterize a given structure. These formulas are based on factors that affect the size of the local scours, that is, their dimensions and location. Examples of such formulas are those contained in this article: *Franke (1960)*, *Straube (1963)*, *Tarajmovič (1966)*, *Rossinski & Kuzmin (1969)* equations. The need to study this phenomenon results from the prevalence of hydrotechnical structures equipped with gates (from small gated checks to large weirs) and from potential damage that may be associated with excessive development of local erosion downstream, including washing of foundations and, consequently, loss of stability of the structure.

**Methods**. This study verifies empirical formulas applied to estimate the geometry parameters of a scour hole on a laboratory model of a structure where water is conducted downstream the gate with bottom reinforcements of various roughness. A specially designed remote-controlled measuring device, equipped with laser scanner, was applied to determine the shape of the sandy bottom. Then the formula optimization is conducted, using Monte Carlo sampling method, followed by verification of field conditions.

**Results**. The suitability of a specially designed device, equipped with laser scanner for measuring the bottom shape in laboratory conditions was demonstrated. Simple formula describing local scour geometry in laboratory conditions was derived basing on the Straube formula. The optimized formula was verified in field conditions giving very good comparative results. Therefore, it can be applied in engineering and designing practices.

Corresponding author
Marta Kiraga,
marta_kiraga@sggw.edu.pl

## INTRODUCTION

The article addresses the subject of hydraulic structures, understood as structures for water management, shaping water resources and water use (*Chen, 2015*). The analyzed case is a damming structure i.e., a structure enabling permanent or periodical maintenance of a water surface level elevation above the adjacent land or body of water. There is a strong link between water and environment, as well as between water and development, both with opposite aims. While the willingness to develop and expand the hydrotechnical infrastructure is a drive for economic development and thus for urbanization to flourish, the environmental aspects direct the designer rather towards sustainable solutions (*Koskinen, Leino & Riipinen, 2008*; *Jordaan, 2009*; *Rasekh, Afshar & Afshar, 2010*) and, as engineering practice demonstrated, to reject a project at an early stage. It should be denoted that a failure of a hydrotechnical structure does not only concern dams of significant sizes but also small hydraulic structures (usually not taken into account by ecologists) which are a potential source of disaster, affecting the life and economy of a given region (*Hossain, 1992*; *Lopardo & Seoane, 2004*).

There are many criteria in the design of hydraulic structures that must be optimized at the same time. Construction, operating and maintenance costs, construction reliability, environmental impact, social disruption and potential loss of life can be identified as one of the most noticeable of a large number of criteria (*Rasekh, Afshar & Afshar, 2010*). Therefore, a comprehensive research on the environmental risk of hydrotechnical structures failure is necessary. One aspect of this type of research is a proper recognition of hydraulic and morphodynamic processes that accompany the construction of water structures using physical and mathematical models (*Lopardo & Seoane, 2004*; *Syvitski et al., 2010*).

Damming up rivers by implementation of a hydraulic structure unavoidably influences the stream course and valley morphology. Upstream the structure, due to the water surface level increment the water can flood the adjacent valley and the reservoir may form. Due to the cross-section enhancement and, therefore, stream velocity reduction, sediment sedimentation and accumulation may occur when the weight of the ground particles will outweigh the transport capacity of the water stream (*Graf, 1998*). Simultaneously, erosion can intensify downstream the structure, especially in the case of very low water surface level and the stream velocity rapid enhancement. The debris-free stream leaving the dammed structure, with additional kinetic energy and increased turbulence, has a high eroding capacity (*Szydłowski & Zima, 2006*; *Pagliara & Kurdistani, 2013*; *Zobeyer et al., 2010*; *Lee & Hong, 2019*). The effect of intensified erosion, which takes place downstream the structure, is mainly local scouring and gradual lowering of the bottom on an increasingly longer section of the river. The increased erosion of a riverbed is unfavorable and undesirable not only due to slow degradation of the riverbed, but also because of occurrence of rapid morphodynamic processes. It is commonly assumed that the most intensive transformation of riverbeds takes place during catastrophic flooding when basic hydrodynamic parameters of the stream increase many times. Excessive development of a scour hole directly behind the structure, such as weir or sluice, poses a threat to its safety as it may lead to washing away the foundation, embankments damages and loss of stability (*Bajkowski, Siwicki &*

*Urbański, 2002*). Removing and repairing these undesirable effects is troublesome and expensive. Therefore, technical solutions are needed to reduce scour hole dimensions.

Due to a focus on ecological changes in aquatic environment and the adjacent area in recent decades, it is worth considering what environmental benefits of local scouring can be. In order to specify them, it is necessary to identify this phenomenon more precisely i.e., to know the causes of its occurrence, its characteristics and the process over time (*Siwicki & Urbański, 2004*).

The specific character of the flow within a scour area is a certain diversity of conditions in comparison with a river in its natural course. Stream velocity and associated physical forces constitute the most important environmental factor affecting organisms living in watercourses (*Cullen, 1991*; *Smith, Goodwin & Nestler, 2014*).

A significant reduction of stream velocity in a bottom local scour area leads to a specific flow region formation named the wall-adjacent boundary layer. This layer can serve as a shelter for organisms from turbulence and high water velocities. In well-developed scour holes, the near-zero stream velocity in the wall-adjacent boundary layer can form local areas of still water (*Lupandin, 2005*; *Liao & Cotel, 2013*; *Hockley et al., 2013*). The adaptation of fish to living in still and flowing water has been extensively studied and it was found that one of the most important environmental factors is the dissolved oxygen content of the water. The higher the stream velocity and turbulence intensity, the more oxygenated the water is. Therefore, the water after passing through the structure is aerated as evidenced by the measured increases in speed and turbulence of the stream in the position downstream compared to the upper structure (*Kobus & Koschitzky, 1991*).

As a result of the process of sorting and armoring, i.e., washing out finer particles and leaving thicker fractions at the bottom, the material is sorted on the scour hole bottom surface and a layer made of thicker fractions is formed. These are the factors that determine the structure of the velocity field at the bottom. This type of scouring process creates quite favourable environmental conditions for many species of fish, i.e., a well aerated stream, thicker material on the bottom and a smooth type of stream in the wall-adjacent boundary layer area (*Siwicki & Urbański, 2004*; *Ochman & Kaszubkiewicz, 2004*; *Hauer et al., 2018*).

Local scouring in the aquatic environment may be particularly desirable during the season of low dischagres. It can then serve as a reservoir in which particular species that require a certain water depth, can survive. Lowering of the river bed may also be beneficial in the construction of a fish pass. The inlet and outlet of fishponds must be submerged deep enough under the water surface level, and this could be difficult in the structure stand downstream, where the depth of the stream is normally lower than in upstream stand area (*Siwicki & Urbański, 2004*). At the design stage, it is important to develop a reliable forecast of a size, shape and position of the local scour, both in dams and small hydraulic structures, such as gated checks, weirs or sluices.

Gated checks are small river training structures, applied for limiting channel incision, bed stabilization, reducing flow velocity and raising upstream water level. These structures are often used in channels where the adjustment of water level is required more frequently or where higher cost compared to stop-logs, are justified (e.g., saving of labour). Gated checks are usually equipped with hand-operated slide gates of various types, from simple

wooden shutters to hand-wheel operated adjustable orifice type gates (*Kraatz & Mahajan, 1975*).

Although many studies on local scouring downstream hydraulic structures can be found in the literature in the recent years (for example *Sun, Wang & Wang, 2012*; *Pagliara et al., 2016*; *Khaple et al., 2017*; *Al-Husseini, Al-Madhhachi & Naser, 2019*; *Singh, Devi & Kumar, 2020*; *Li et al., 2020*; *Taha et al., 2020*; *Wang et al., 2020*; *Yan, Rennie & Mohammadian, 2020*), among them only a few focus on small hydraulic structures (*Lopardo & Seoane, 2004*; *Kiraga & Popek, 2016*; *Odgaard, 2017*; *AL-Suhaili, Abbood & Samir Saleh, 2017*; *Kiraga & Popek, 2018*). Gated checks in particular are a rare object of research, which, given quite high prevalence of this type of structures, determines the appropriateness of the undertaken subject.

In recent years, the role of small retention of valley areas was emphasized (*Boix et al., 2012*; *Mioduszewski, 2014*). A significant role in maintaining and increasing retention is played by small water structures (*Lopardo & Seoane, 2004*) which are less popular in the research subject matter than bridges or larger weirs with controlled closures.

It should also be noted that hydraulic instrumentation has advanced significantly, so fundamental flow parameters or river bed shape can be measured with greater precision. For instance, the PIV imaging anemometry system can be used to describe the distribution of velocity fields in the area of water structure and the description of a bottom shape can be perform using an echo sounder, e.g., fixed on a boat, or by laser scanning technique (*Hager & Boes, 2014*; *Killinger, 2014*).

Most of the research samples in order to identify the factors influencing the amount of local scouring to the greatest extent below water structures were carried out in laboratory conditions. The following factors that influence a scour hole shape and location are:

–related to the flume geometry (width, depth, bed inclination);

–related to the type and geometry of the structure (type of structure, reinforcement construction, dimensions of upstream and downstream part of the structure elements);

–characterizing water flow conditions (flow rate, average speed, hydraulic gradient, bed shear stress, flow resistance),

–water properties (density, viscosity),

–characterizing the flume material (grain size, grain distribution, density, porosity, roughness),

–characterizing the conditions of sediment transport (critical velocity, critical shear stress, sediment transport intensity),

–time (*Graf, 1998*; *Breusers & Raudkivi, 1991*; *Kiraga & Popek, 2019*).

In spite of many experimental works carried out under various constructional conditions and high variability of hydraulic conditions, the universal principles of calculating the local scour dimensions and transferring it to field conditions are still unknown. Solutions are sought, involving different coefficients, which characterize a given construction, based on identified factors that influence scour size and position (*Franke, 1960*; *Straube, 1963*; *Tarajmovič, 1966*; *Rossinski & Kuzmin, 1969*). The formation and expansion of local scouring that results from time-varying, two-phase movement of water and sediment is one of the most undiscovered processes in hydrotechnical engineering (*Graf, 1998*;

*Nouri Imamzadehei et al., 2016*). Despite numerous studies carried out since the first decades of last century (for example *Lacey, 1946*; *Ahmad, 1953*; *Breusers & Raudkivi, 1991*; *Lenzi, Marion & Comiti, 2003*; *Ślizowski & Radecki-Pawlik, 2003*; *Ben Meftah & Mossa, 2006*; *Kiraga & Popek, 2016*; *Pagliara et al., 2016*; *Kiraga & Popek, 2018*; *Al-Husseini, Al-Madhhachi & Naser, 2019*), there is no sufficient and unquestionable basis for the mathematical description of the process of local erosion, and thus for a development of forecasts of scour holes that will occur during the design of structures. Also, it is not always possible to predict fully reliable estimation based on the results of laboratory tests, because in laboratories the researchers are usually unable to lead to the occurrence of the so-called final scour, i.e., to a state in which the extension of the duration of the experiment does not cause changes in the dimensions and location of sandy bottom and banks (*Chabert & Engeldinger, 1956*; *Barbhuiya & Dey, 2004*). Moreover, designers find it difficult to choose those that give reliable results. Due to the diversity of applied constructions of structures and the variability of hydraulic conditions, it is difficult to generalize the derived formulas ((*Graf, 1998*; *Barbhuiya & Dey, 2004*; *Ben Meftah & Mossa 2006*).

The absence of a forecast of the effects of local erosion makes it impossible to rationally assess the degree of safety and certainty of the use of a hydraulic structure, since the scour poses a similar threat to the structure, such as insufficient structure capacity, too low structural stability coefficient, ground strength exceeding, etc.

The complexity of a local scouring process means that only fragments of the problem are usually examined with limited objectives, such as:

–explanation of the influence of factors, e.g., the construction of downstream part of the structure, length and roughness of the reinforcements, etc. (*Rossinski & Kuzmin, 1969*; *Al-Mohammed, Jassim & Abbass, 2019*);

–highlighting the effectiveness of various design solutions to prevent excessive erosion (*Epely-Chauvin, DeCesare & Schwindt, 2014*; *Taha et al., 2020*);

–formulation of dependencies, formulas, etc. to determine scour forecasting for assumed geometrical, hydraulic and ground conditions (*Gaudio & Marion, 2003*; *Kiraga & Popek, 2019*).

Additionally, the results of tests carried out in a laboratory are difficult to translate directly into field conditions due to the scale effect (*Farhoudi & Smith, 1985*), whereas during field tests problems result mainly from the lack of knowledge of the initial conditions, i.e., the shape of the bottom before disturbing the existing dynamic balance in the channel (*Lenzi, Marion & Comiti, 2003*; *Pagliara et al., 2016*).

Researchers agree that regardless of a construction of a hydraulic structure, the depth of the scour hole is influenced by length, roughness and height position of the fortifications downstream (*Rossinski & Kuzmin, 1969*; *Urbański, 2008*; *Taha et al., 2020*). Difficulties in explaining and presenting the influence of factors on the process of scouring and the lack of perspectives for establishing universal relations between the complicated flow system and the sediment transport, lead to use simple, in fact intuitive, relationships that allow to determine the depth of local scour.

The estimation of the maximal scour hole depth and the channel reach infested by extensive erosion allows for a proper design of the downstream of hydraulic structure,

ensuring safety and stability, as well as reducing the construction and subsequent operation cost. Therefore, the estimation of the geometry of forecasted scour should be an integral part of the design stage of hydrotechnical structures (*Brandimarte, Paron & Di Baldasarre, 2012*; *Prendergast & Gavin, 2014*).

Difficulties of local scouring investigations result primarily from the multitude of factors influencing its shape and dimensions. The following factors can be mentioned among them (*Franke, 1960*; *Straube, 1963*; *Tarajmovič, 1966*; *Rossinski & Kuzmin, 1969*):

–relation to the flume or channel geometry (e.g., the Shalash and Franke, Müller, Tajarmovič formula);

–relation to the type and geometry of the structure (e.g., the Rossinski formula);

–relation to water flow conditions, such as flow rate, average speed or flow resistance);

–water physical properties;

–relation to bed material (e.g., the Straube method).

This paper comprises the identification, verification and validation of chosen empirical formulas: Shalash & Franke, Straube, Müller and Tarajmovič serving to estimate the scour dimensions in local scour process forming due to damming up the flume by the gate, equipped with downstream stage embankment. For formula optimization the Monte Carlo sampling method was applied. Laboratory research was performed as a first part of the studies, then the formula best describing flume experiment was verified in field conditions.

## MATERIALS & METHODS

Research based on laboratory studies was performed in a 11-m long hydraulic flume with 0.58-m width. No bed inclination downstream was introduced, however it should be noted that lowland rivers that formed in alluvial depositions usually have gradients of 0.5–3%. If such a slope were to be reproduced in the present laboratory conditions, the difference in elevation of the bottom below the water structure would be one mm to a maximum of one cm.

Data were collected as previously described in the publication "Bed shear stress influence on local scour geometry properties in various flume development conditions" in *Water* by *Kiraga & Popek (2019)*. Specifically, the research assumed bed shape measurements during local scouring formation, both using pin gauge, laser scanning of the surface and water surface level examination in presumed hydraulic conditions. However, the examined flume development differs from the mentioned publication in *Water*. Namely, two gated check models assumed slide gate introduction, which was constantly raised to five cm of height to ensure invariable flow area of 0.029 m² (Figs. 1A, 1B). In the vicinity of the damming structure, the bottom was solid on the length of $L_1 = 0.30$ m upstream the gate and $L_2 = 0.80$ m downstream, designed to imitate bed reinforcement typical for weirs or other river training structures, often made of concrete or rip rap. The reinforcement downstream the check was made of plain slab working as a reinforcement within the Model I of flume development (Figs. 2A, 3A) (4), whereas Model II assumed stone riprap reinforcement made of rocks (8) whose medium height was 1.5 cm (Figs. 2B, 3B). A scour hole formed inside a sandy part downstream the check with a length $L_3$ of 2.20 m (2). The

applied constructional solution of the model assumed representation of water flow under a partially open valve of the gated check with lowered reinforced bottom below with variable roughness where sediment transport takes place through the gate—not held by any weir. According to *Kraatz & Mahajan (1975)* the length of the reinforcement downstream the gated check should be ca. 1.5 times longer than the width of the gate—and the model applied within present experiment ensure the elements' dimensions close to this ratio.

Bottom shape was investigated at all flume lengths. After each measurement series, the sand in the flume was dried for ca. 13 h - the outflowing water was removed by using drainage pipe (5 in Figs. 2A and 2B). The measurement schedule was similar as previously described in "Bed shear stress influence on local scour geometry properties in various flume development conditions" in *Water* by *Kiraga & Popek (2019)*. Namely, pin water gauges were used in order to measure water surface elevation at the intake part and along the flume in the central axis (1). The water surface level was regulated with an outlet gate (6). Before introducing water into the flume and after draining the sand the final level of sandy bottom was measured with a laser scanner (7) and with a moving disc probe (1) as a supportive device in presumed time steps (0.5–2 h).

Before starting each measurement series and introducing water stream into the flume, the sandy bed was uniformly adjusted to a constant level and compacted with load of 2.5 kg dropped to the bed surface with an energy of about 5 J. Then, the position of the bottom was measured with a disc probe with presumed mesh 5 ×7 cm to 20 ×10 cm both upstream and downstream the hydraulic structure.

As described in "Bed shear stress influence on local scour geometry properties in various flume development conditions" in *Water* by *Kiraga & Popek (2019)* flume side walls were made of glass with a roughness coefficient $n_w = 0.010$ s m$^{-1/3}$. The soil used during the study was uniform coarse sand with medium diameter $d_{50} = 0.91$ and $d_{95} = 1.2$ mm and roughness coefficient $n_b = 0.028$ s m$^{-1/3}$. Experiments were performed in the scope of steady water flow discharge within the following range $Q_w = 0.010 - 0.045$ m$^3$ s, water depth downstream the structure $h = 0.05 - 0.26$ m and Froude number $Fr < 1$. 29 measurement series were performed, each lasting 8 h (9 measurement series on model I and 20 on model II) (Tables 1 and 2).

No sediment feeding system was adopted. Bedload transport conditions were assured by specific set of hydraulic conditions that invokes particle movement from upstream towards downstream. Therefore, the experiment was carried out in 'live-bed' conditions, where soil leaving the scour hole is substituted by approaching load from the upstream. It is worthy to notice that for typical lowland rivers both bedload load and suspended load are present in various relations. For example, the suspended load constitutes 60–70% of the whole sediment load transported by Vistula River in Poland and 50–90% of that is transported by its tributaries (*Lajczak, 1996*), although only bedload was investigated in this study.

A group of experiments carried out in a hydraulic laboratory, the results of which were published, among others, in *Water* or *IEEE Access* journals (*Kiraga & Popek, 2016*; *Kiraga et al., 2018*; *Kiraga & Popek, 2019*; *Kiraga & Miszkowska, 2020*), concerned the phenomenon of formation of local scouring as a result of not only varied roughness of the materials building the bed, but also the restriction of the flow field by inserting a model of

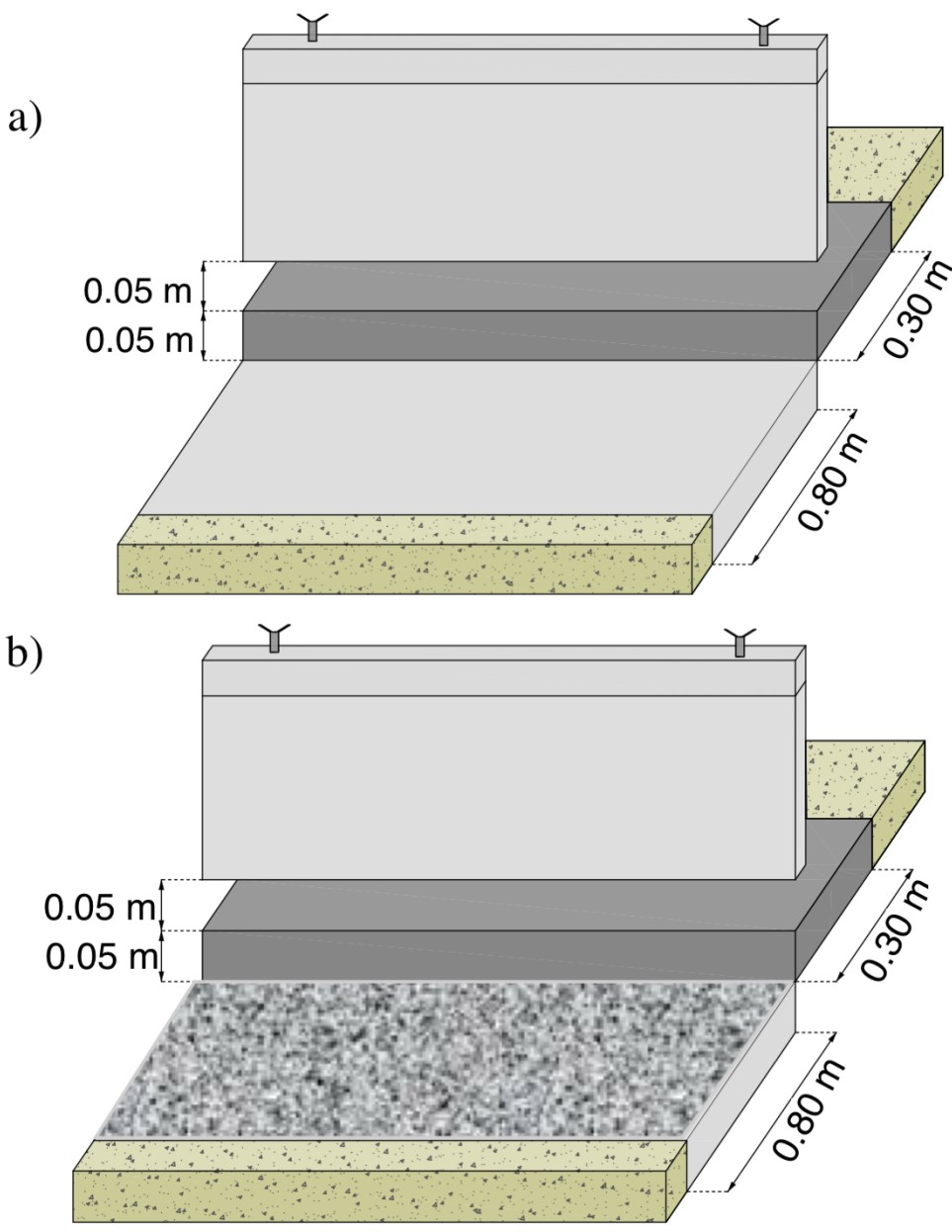

**Figure 1** **Gated check details.** (A) Model I (gated check without additional roughness downstream); (B) model II (gated check with additional roughness downstream).

damming structures into the flume. Namely, due to the flow resistance increment along the whole flume resulting from varied roughness of solid and sandy bottom, the hydraulic gradient increases causing the intensification of shear stress at the bottom. After exceeding the critical shear stress, the motion of sediment grains starts, followed by gradual scouring of the bed. Maximal scour depth $z_{max}$, scour length $L_s$ and the distance between the deepest point of the hole and the end of reinforcement $L_e$ were examined (Fig. 4) during each measurement.

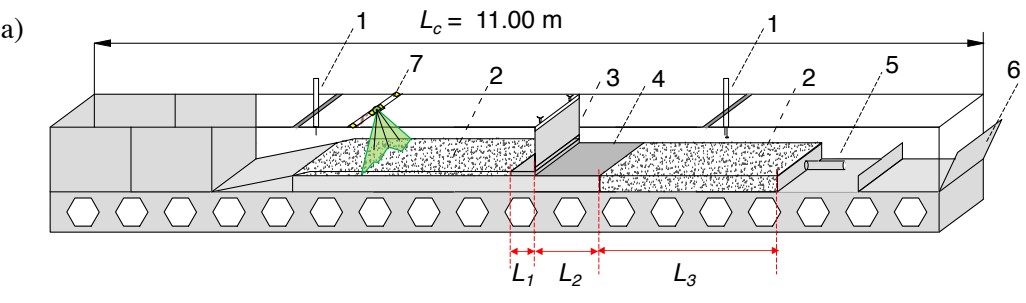

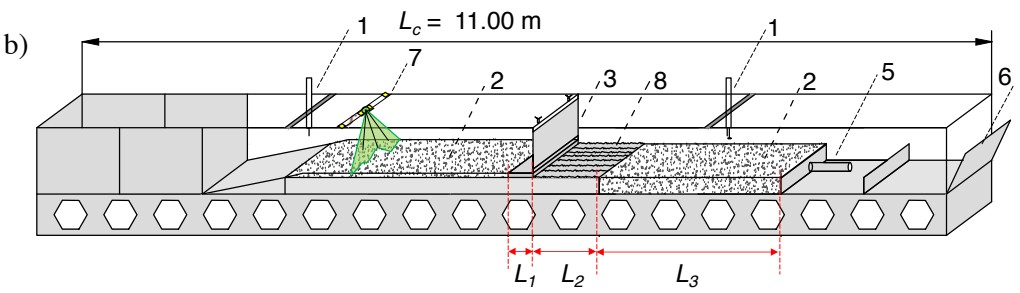

**Figure 2   Laboratory test stand.** (A) Model I (gated check without additional roughness); (B) Model II (gated check with additional roughness), where: 1–movable pin gauge equipped with disc probe; 2–sandy bed; 3–gate; 4–reinforcement; 5–drainage; 6–outlet gate; 7–laser scanner; 8–reinforcement with additional roughness.

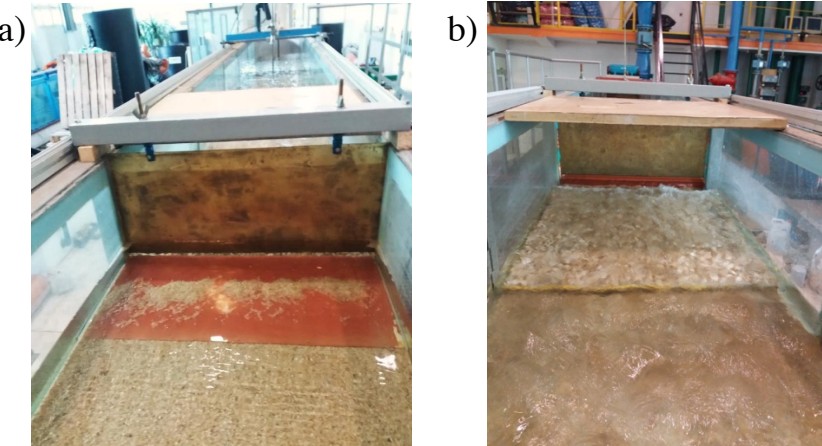

**Figure 3   Photography of laboratory model.** (A) Model I; (B) model II.

In order to investigate the final scour shape both a device equipped with laser scanner and a disc probe were applied as supportive devices. Data were collected as previously described in "Bed shear stress influence on local scour geometry properties in various flume development conditions" in *Water* by *Kiraga & Popek (2019)*. Prototype A1 of the device was engineered in 2016 by Marta Kiraga and Matvey Razumnik within the

**Table 1  Model I—measurement series summary table.** Where: $Q_w$, water flow discharge; $h$, initial water depth in control profile; $T$, measurement duration; $Fr$, Froude number.

| No. of measurement series | $Q_w$ [m³s⁻¹] | $h$ [m] | $T$ [s] | $Fr$ [-] |
|---|---|---|---|---|
| 1 | 0.025 | 0.13 | 28800 | 0.31 |
| 2 | 0.020 | 0.05 | 28800 | 0.98 |
| 3 | 0.023 | 0.10 | 28800 | 0.40 |
| 4 | 0.030 | 0.08 | 28800 | 0.73 |
| 5 | 0.025 | 0.05 | 28800 | 0.99 |
| 6 | 0.030 | 0.11 | 28800 | 0.45 |
| 7 | 0.028 | 0.11 | 28800 | 0.42 |
| 8 | 0.026 | 0.10 | 28800 | 0.45 |
| 9 | 0.029 | 0.08 | 28800 | 0.71 |

**Table 2  Model II—measurement series summary table.** Where: $Q_w$, water flow discharge; $h$, initial water depth in control profile; $T$, measurement duration; $Fr$, Froude number.

| No. of measurement series | $Q_w$ [m³s⁻¹] | $h$ [m] | $T$ [s] | $Fr$ [-] |
|---|---|---|---|---|
| 1 | 0.020 | 0.05 | 28800 | 0.98 |
| 2 | 0.023 | 0.10 | 28800 | 0.40 |
| 3 | 0.030 | 0.08 | 28800 | 0.73 |
| 4 | 0.025 | 0.06 | 28800 | 0.99 |
| 5 | 0.030 | 0.11 | 28800 | 0.45 |
| 6 | 0.028 | 0.11 | 28800 | 0.42 |
| 7 | 0.026 | 0.10 | 28800 | 0.45 |
| 8 | 0.029 | 0.08 | 28800 | 0.71 |
| 9 | 0.024 | 0.08 | 28800 | 0.58 |
| 10 | 0.029 | 0.10 | 28800 | 0.50 |
| 11 | 0.013 | 0.06 | 28800 | 0.48 |
| 12 | 0.013 | 0.06 | 28800 | 0.57 |
| 13 | 0.014 | 0.06 | 28800 | 0.52 |
| 14 | 0.021 | 0.06 | 28800 | 0.78 |
| 15 | 0.022 | 0.09 | 28800 | 0.46 |
| 16 | 0.022 | 0.07 | 28800 | 0.65 |
| 17 | 0.027 | 0.07 | 28800 | 0.80 |
| 18 | 0.028 | 0.08 | 28800 | 0.67 |
| 19 | 0.024 | 0.07 | 28800 | 0.72 |
| 20 | 0.030 | 0.07 | 28800 | 0.88 |

university grant for young researchers "The influence of small hydraulic structures on sediment transport conditions" (*Kiraga & Popek, 2018*).

Prototype A1 (Fig. 5) is equipped with a laser rangefinder and automatic movement system embedded on guides along the flume, scanning the bottom area with a demanded grid (every one mm in case of present experiments). Grid density alteration possibility gives a far greater accuracy of measurement than the disc probe. The use of the device ensures

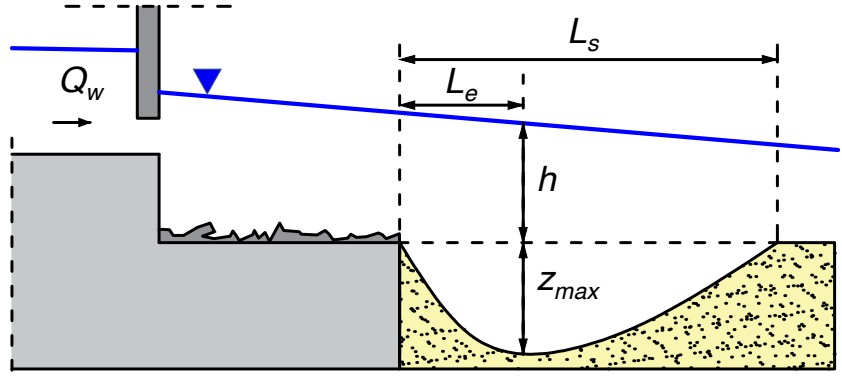

**Figure 4  Local scour geometrical parameters.** $z\_max$, maximal scour depth; h, water depth before scour formation; $L_s$, scour length; $L_e$, the distance between the deepest point of the hole and the end of reinforcement.

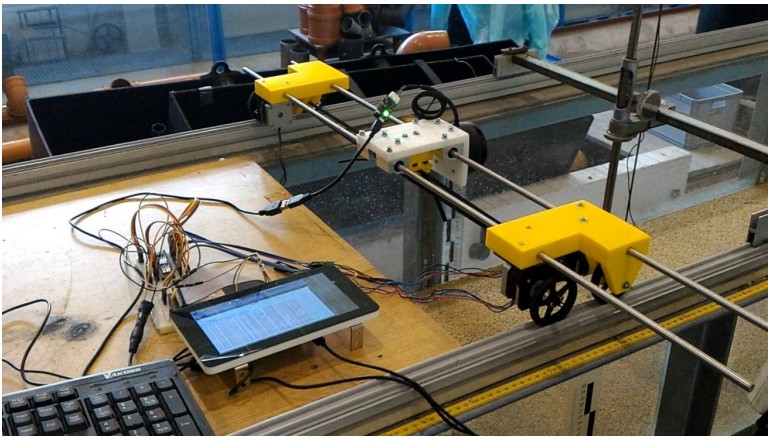

**Figure 5  Prototype A1 during laboratory measurements.**

data transmission directly in digital form, so that the coordinates can be easily processed to obtain the desired scour hole geometrical parameters.

Laser scanning, also known as LiDAR (Light Detection and Ranging) is an active tele-detection method which uses electromagnetic waves sent by the emitter. The result is point cloud with coordinates (*x, y, z*) (*Jaboyedoff et al., 2010*; *Killinger, 2014*). The measuring system (LiDAR) consists mainly of a transmitter i.e., a module generating laser light (diodes), an optical telescope focusing the returning reflected radiation and a detector converting light energy into an impulse recorded in the module that records the acquired data.

The prototype's supporting elements are made of biodegradable polyactide (PLA) and are printed on a 3D printer. Using Raspberry Pi microcomputer allows simultaneous computations and data collecting by the beam. The device is fully automated which was

achieved by the application of a single board computer, dedicated software and the set of stepper motors, which results in measurements repeatability, constant accuracy on demand and fast execution of results. The obtained coordinates mesh is characterized by high resolution: one mm by one mm–therefore bottom shape is described very precisely, both in numerical form and as a graphical tracing. Numerical cloud can be easily transformed thence scour hole dimensions such as length or depth can be estimated. LiDAR technology application in scour shape and its volume in flume experiments is based on the introduction of an automatic measuring module which, placed above the bottom on a specially prepared controllable system of guides, describes its shape by creating a point cloud.

Deriving from a statement that scouring process stops when stream velocity $v$ is equal to non-scouring velocity $v_n$ Rossinski (*Rossinski & Kuzmin, 1969*) stated that water depth above the local scour can be calculated as:

$$H = z_{max} + h = k_1 \sqrt[1,2]{q/v_{n1}} \tag{1}$$

where $z_{max}$ is local scour depth, [m]; $h$ is water elevation before scour formation (See Fig. 4), [m]; $k_1$ is a dimensionless coefficient, describing intensified turbulence of the stream, [-]; $q$ is unit discharge, [m$^3$ s$^{-1}$ m$^{-1}$]; $v_{n1}$ is non-scouring velocity for water depth of 1 m, depending to soil properties, [m$^2$ s$^{-1}$] calculated as following:

$$v_{n1} = \sqrt{2g(\gamma_r - \gamma_w)/1.75\gamma_w d_{50} log(8.8/d_{95})} \tag{2}$$

in which $g$ is gravity acceleration, $g = 9.81$ m s$^{-2}$; $\gamma_r$ and $\gamma_w$ are specific weights, sediment and water, respectively [N m$^{-3}$]; $d_{50}$ and $d_{95}$ are diameters that correspond to 50% and 95% of particles finer than the reported particle size.

The $k_1$ value in the formula (Eq. 1) is an empirical coefficient, dependent on downstream development conditions. Based on practice experiences $k_1$ takes the value of 1.70 when the reinforcement downstream the gate is not deepened and sheet piling, palisade or other vertical securing element make an additional protection. Due to the stream energy enhancement in the region of the gate outlet without any energy dissipating device local scouring process is intensified. When transverse trench is dug downstream the reinforcement, of depth equal to the expected depth of the scour; and the slope of this trench is no more than 1:4, then $k_1 = 1.05$ should be assumed (Figs. 6A, 6B).

Experimental case is referred to the conditions when coarse sandy bed is preceded by deepened reinforcement downstream (Fig. 6C) therefore, empirical studies on $k_1$ parameter were needed.

The difficulty of explaining and presenting the impact of factors influencing local scouring process in large scale hydraulic structures with the lack of perspectives for establishing relations between a complicated flow system and sediment transport, forces to apply simple, intuitive relations allowing for the determination of the depth of scour holes. Scour length $L_s$ and the distance between the deepest point of the hole and the end of reinforcement $L_e$ where the stream comes out from under the gate were determined by several authors:

–According to Shalash and Franke (*Franke, 1960*):

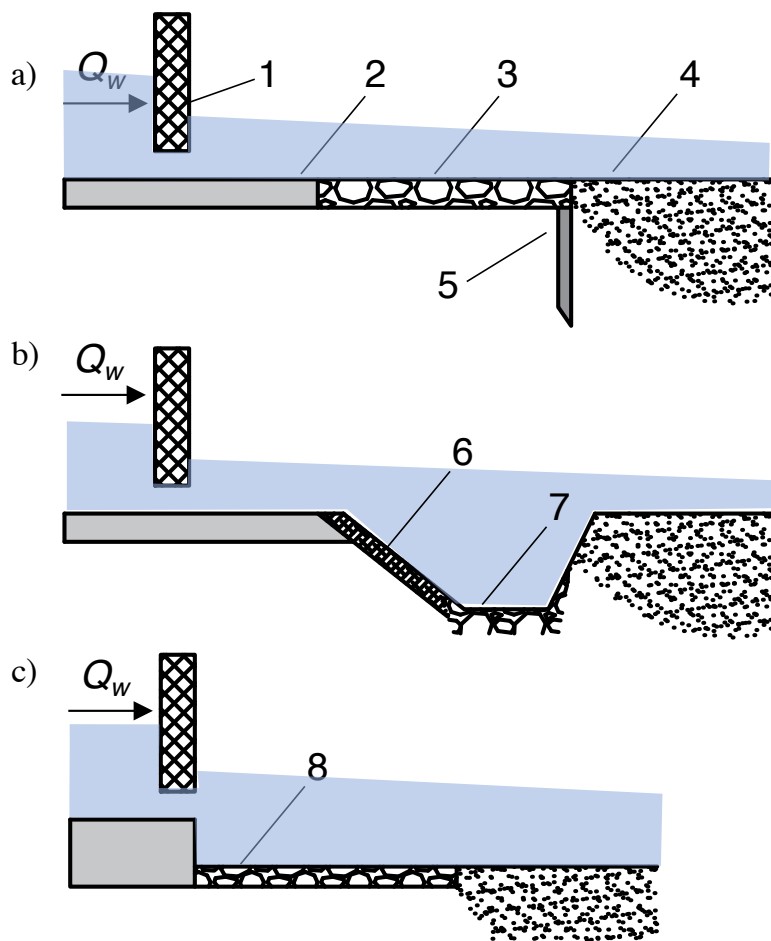

**Figure 6 Various types of lower stage of gated check construction.** (A) Reinforcement equipped with sheet piling, (B) reinforcement followed by transversal trench, (C) sandy bed following reinforcement with lowered bottom, where: 1–gate; 2–solid bottom; 3–stone reinforcement; 4–sandy bed; 5–sheet pilling or palisade; 6–bank reinforcement; 7–transverse trench with stone bottom; 8–lowered reinforcement.

$$L_s \text{ and } L_e = f(z_{max}) \tag{3}$$

$$L_s = 11 \cdot z_{max} \tag{4}$$

$$L_e = 6.6 \cdot z_{max} \tag{5}$$

–According to Müller (*Franke, 1960*; *Straube, 1963*):

$$L_s \text{ and } L_e = f(z_{max}) \tag{6}$$

$$L_s = (9.9 \div 0.8) \cdot z_{max} \tag{7}$$

$$L_e = (4.9 \div 0.5) \cdot z_{max} \tag{8}$$

or

$$L_s \text{ and } L_e = f(z_{max}, h) \tag{9}$$

$$L_s = (6.0 \div 1.22) \cdot (z_{max} + h) \tag{10}$$

$$L_e = (2.94 \div 0.59) \cdot (z_{max} + h) \tag{11}$$

–According to Straube (*Straube, 1963*):

$$L_s = f(z_{max}, q, h, d_{50}) \tag{12}$$

$$L_s = 8.0q^{0.36}(z_{max} + h)d_{50}^{-0.14}h^{-0.40} \tag{13}$$

$$L_e = f(L_s, h, d_{50}) \tag{14}$$

$$L_e = 0.39L_s d_{50}^{0.12}h^{-0.12} \tag{15}$$

The formulas (Eqs. 3–15) are recommended for systems in which the outflow from the gate goes directly onto unreinforced ground. For constructions equipped with reinforcement the Tajarmovič formula is recommended (*Tarajmovič, 1966*):

$$L_e = f(z_{max}) \tag{16}$$

$$L_e = 12.75z_{max}^{0.5} \tag{17}$$

Monte-Carlo integration works by comparing random samples with the function value. Straube equations can be described generally in the following forms:

$$L_s = aq^b(z_{max} + h)d_{50}^{-c}h^{-d} \tag{18}$$

$$L_e = kL_s d_{50}^{m}h^{-p} \tag{19}$$

where $a$, $b$, $c$, $d$, $k$, $m$, $p$ are function parameters which were sampled in the following ranges:
  $-a \in \;<5.0, 8.0>$;
  $-b \in \;<0.24, 0.40>$;
  $-c \in \;<0.10, 0.20>$;
  $-d \in \;<0.35, 0.45>$;
  $-k \in \;<0.30, 0.60$;
  $-m \in \;<0.01, 0.13>$;
  $-p \in \;<0.01, 0.20>$.
Using a random number generator in the assumed ranges of values, 6000 combinations of parameters $a$, $b$, $c$, $d$ for Eq. (18) and 6000 combinations of parameters $k$, $m$, $p$ for

**Table 3  Model I—scour geometry parameters summary table.** Where: $q$, unit water flow discharge; $z_{max}$, maximal scour depth; $Le$, the distance between the deepest point of the hole and the end of reinforcement; $L_s$, scour length.

| No of measurement series | $q$ $[\text{m}^3\text{s}^{-1}\text{m}^{-1}]$ | $z_{max}$ [m] | $H_{max}$ [m] | $L_e$ [m] | $L_s$ [m] |
|---|---|---|---|---|---|
| 1 | 0.0431 | 0.0201 | 0.1451 | 0.59 | 2.10 |
| 2 | 0.0345 | 0.0911 | 0.1411 | 0.66 | 2.10 |
| 3 | 0.0397 | 0.0532 | 0.1532 | 0.68 | 2.20 |
| 4 | 0.0517 | 0.0821 | 0.1621 | 0.67 | 2.18 |
| 5 | 0.0431 | 0.1020 | 0.1600 | 0.78 | 2.01 |
| 6 | 0.0517 | 0.0672 | 0.1772 | 0.78 | 2.18 |
| 7 | 0.0483 | 0.0511 | 0.1611 | 0.76 | 2.20 |
| 8 | 0.0448 | 0.0630 | 0.1630 | 0.66 | 2.20 |
| 9 | 0.0500 | 0.0772 | 0.1572 | 0.71 | 2.20 |

Eq. (19) were selected. The average relative error $\delta$ for all 29 series of measurements was chosen as a criterion for evaluation of the formula described by a given combination of parameters.

The key to the accuracy and correctness of the Monte Carlo method is a random number generator. The method presents a solution to a problem as a parameter of a hypothetical population. Using a sequence of random numbers, it creates a population sample from which estimated values of the sought parameters can be obtained (*Niederreiter, 1992*).

Next step was to verify the optimized formula on independent measurement results published in 2010 by Gaudio and Marion, performed on a flume with sandy bottom with the hydraulic structure represented by the cascade of transversal sills.

## RESULTS

Basic geometric parameters of observed scour during 29 measurement series, each characterized by unit discharge $q$ (9 on Model I and 20 on Model II) are presented in Tables 3 and 4. Non-scouring velocity for water depth of 1 m $v_{n1}$ in presumed grain conditions was equal to 0.502 m s$^{-1}$. Maximal scour depth ranged from 1 to 10 cm. The criterion for the reach infested by the scour is bed level, i.e., scour, is recognized within an area in which the depth of the bottom after 8-hour measurement series exceeds 10% of the maximum hole depth (*Kiraga & Miszkowska, 2020*).

Within the scope of the assumed measurement schedule each measurement series were carried out with unique, within each model, variable combinations of input flow rate and water level. However, by means of variability of those combinations, the same values of unit flow rate $q$ (per unit width) were obtained, which leads to a conclusion that in respect of unit flows, the repeatability of the experiments was ensured. Moreover, during the laboratory tests it was necessary to repeat some measurement series several times, e.g., due to faulty transfer of numerical data from the microcomputer used, which made it possible to check the repeatability of test results. The repetition of the tests was performed assuming the measurement series duration and under the same hydraulic conditions. Differences in

**Table 4** **Model II—scour geometry parameters summary table.** Where: $q$, unit water flow discharge; $z_{max}$, maximal scour depth; $L_e$, the distance between the deepest point of the hole and the end of reinforcement, $L_s$, scour length.

| No of measurement series | $q$ $[m^3 s^{-1} m^{-1}]$ | $z_{max}$ $[m]$ | $H_{max}$ $[m]$ | $L_e$ $[m]$ | $L_s$ $[m]$ |
|---|---|---|---|---|---|
| 1 | 0.0345 | 0.0700 | 0.1200 | 0.80 | 2.19 |
| 2 | 0.0397 | 0.0143 | 0.1143 | 0.56 | 2.20 |
| 3 | 0.0517 | 0.0287 | 0.1087 | 0.61 | 2.20 |
| 4 | 0.0431 | 0.1020 | 0.1600 | 0.61 | 2.20 |
| 5 | 0.0517 | 0.0487 | 0.1587 | 0.79 | 2.20 |
| 6 | 0.0483 | 0.0610 | 0.1710 | 0.27 | 2.20 |
| 7 | 0.0448 | 0.0313 | 0.1313 | 0.79 | 2.20 |
| 8 | 0.0500 | 0.0412 | 0.1212 | 0.63 | 2.20 |
| 9 | 0.0414 | 0.0410 | 0.1210 | 0.31 | 1.77 |
| 10 | 0.0500 | 0.0175 | 0.1175 | 0.58 | 2.20 |
| 11 | 0.0220 | 0.0220 | 0.0820 | 0.33 | 1.00 |
| 12 | 0.0231 | 0.0321 | 0.0871 | 0.41 | 1.10 |
| 13 | 0.0240 | 0.0430 | 0.1030 | 0.51 | 1.50 |
| 14 | 0.0360 | 0.0673 | 0.1273 | 0.53 | 1.98 |
| 15 | 0.0385 | 0.0244 | 0.1144 | 0.51 | 1.60 |
| 16 | 0.0375 | 0.0873 | 0.1573 | 0.66 | 2.20 |
| 17 | 0.0465 | 0.0530 | 0.1230 | 0.61 | 2.20 |
| 18 | 0.0475 | 0.0510 | 0.1310 | 0.56 | 2.13 |
| 19 | 0.0415 | 0.0511 | 0.1211 | 0.55 | 2.01 |
| 20 | 0.0510 | 0.0271 | 0.0971 | 0.67 | 1.70 |

the bottom formation were shown, described by means of basic geometrical parameters of the scouring in the range of 0.3–1.9% in the maximum depth of the scour hole and 2.2–4% in the range of the average depth of the scour, which indicates high repeatability of test results (Figs. 7A, 7B, 7C). Slightly more significant deviation was connected with scour parameters connected with its length: relative error ranging within 0.5–13.6% was met in total scour length and 3.0–16.6% in the case of the distance from the end of reinforcement to the deepest scour point.

The Rossinski formula (Eq. 1) parameters were identified for investigated test stand due to lack of the present gate check structure construction analyses so far. Parameters identification was performed on the basis of mean relative error $\delta$ between observed scour depth and calculations results (Parameter $k_1$ was tested in the range 0.00 to 2.00. With $k_1$ equal to 1.10, the mean relative error reached the minimum value. For the entire tested range of $k$ 1 values, errors in the range of 15–100% were achieved (Fig. 8).

Parameters of 4,5; 7,8; 10,11; 13; 15 and 17 formulas were verified for two models of gated check development. Calculated parameters of observed scour were examined in comparison with the measured ones. The criterion of comparison evaluation was mean

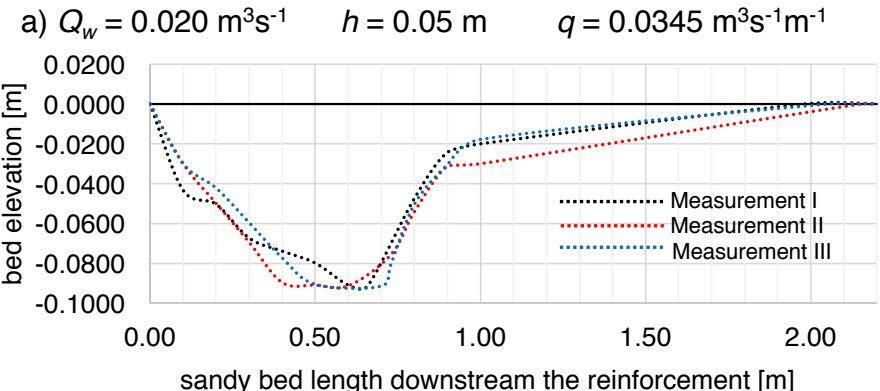

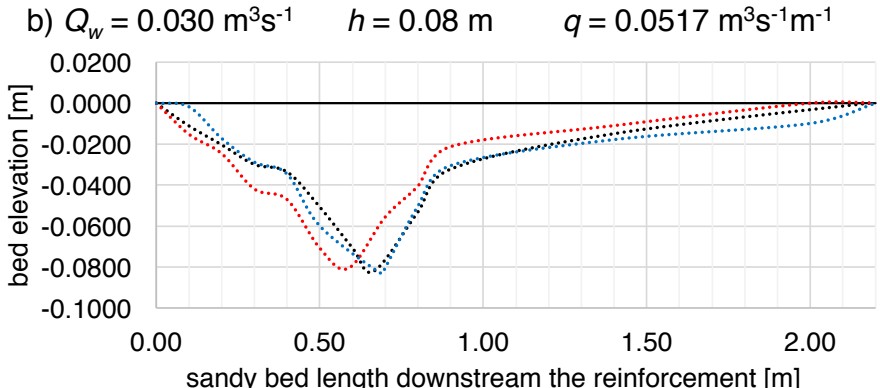

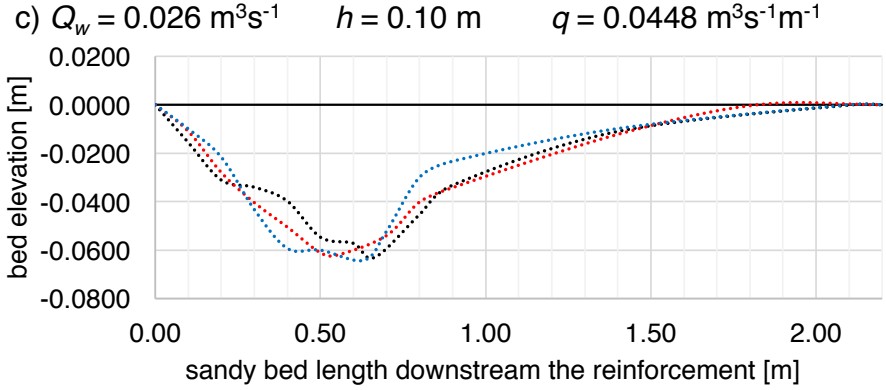

**Figure 7 Shape of the local scour downstream of the bed reinforcement in the Model II flume development during (A–C) additional supplementary measurements, where: (A) unit water discharge $q =$ 0.0345 m³ s⁻¹ m⁻¹; (B) unit water discharge $q = 0.0517$ m³ s⁻¹ m⁻¹; (D) unit water discharge $q = 0.0448$ m³ s⁻¹ m⁻¹.**

relative error of each scour parameter estimation $\delta$ (Table 5) calculated for each group of 29 measurements.

The limitation in determining the range of a scour hole was the length of sandy part (bottom edge) $L_3$, which was 2.20 m. In field studies, the length is long enough for the full
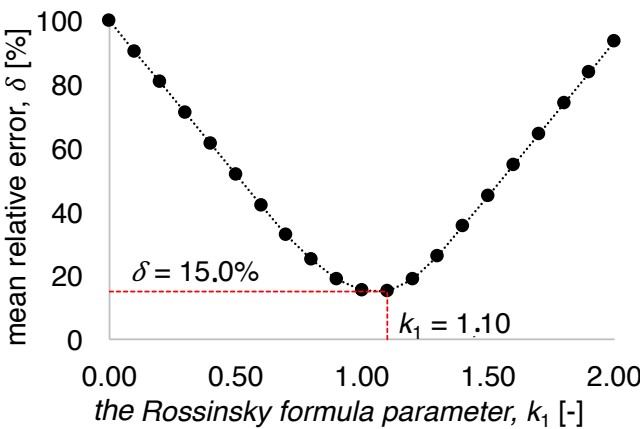

**Figure 8** $k_1$ coefficient impact on the mean relative error of calculations using Rossinski formula (**Eq. (1)**).

**Table 5** Formulas verification summary table. Where: (1–17), number of formula; $d_{Hmax}$, an error of the depth of the water above the deepest point of the scour calculation; $d_{Ls}$, an error of the scour length calculation; $d_{Le}$, an error of the distance between the deepest point of the scour and the end of reinforcement calculation.

| Author's Name | $d_{Hmax}$ [%] | $d_{Ls}$ [%] | $d_{Le}$ [%] |
|---|---|---|---|
| Rossinski | (1) 15.0% | | |
| Shalash & Franke | | (4) 69.8% | (5) 45.2% |
| Müller | | (7) 72.8% | (8) 57.0% |
| Müller | | (10) 56.9% | (11) 38.3% |
| Straube | | (13) 34.2% | (15) 32.1% |
| (Optimized) Straube | | (20) 10.1% | (21) 18.2% |
| Tajarmovič | | | (17) 392.7% |

scour length development –there is no sandy bed length limitation. As mentioned above, the field of the scour was considered to be an area where the bottom lowering exceeded 10% of its maximum depth in presumed time step (Fig. 9). If another criterion was to be adopted, for example, a consideration of the scour hole area within the region where a bottom lowering exceeds 15 or more % of the maximum scour depth, a limiting effect of the sandy bottom downstream the structure could be avoided in some measurement series.

Investigated equations were divided into two groups:

–simple formulas based only on maximal local scour depth $z_{max}$ or on $z_{max}$ and water depth $h$ before scour formation (Eqs. (4), (7) and (10) for total scour length estimation and Eqs. (5), (8), (11) and (17) for the distance between the deepest point of scour and the end of reinforcement estimation).

–formulas involving not only local scour hole depth $z_{max}$ and the depth of water above the unwashed bottom $h$, but also grain characteristics, represented by $d_{50}$ diameter and hydraulic parameter, i.e., unit water discharge $q$ (the Straube formula –Eqs. (13) and (15))

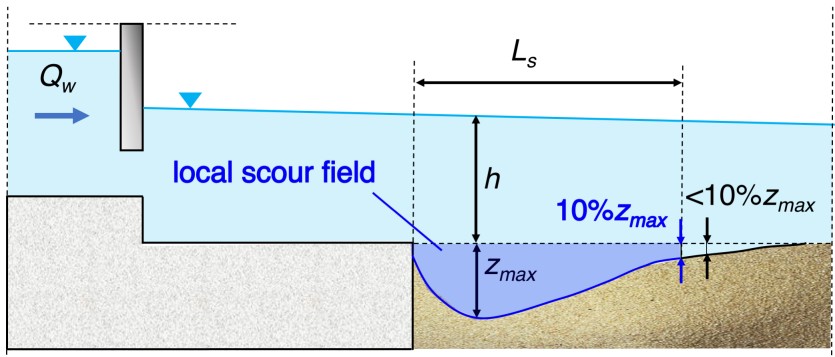

**Figure 9 Local scour field characteristics.** $z_{max}$, maximal scour depth; h, water depth before scour formation; $L_s$, scour length.

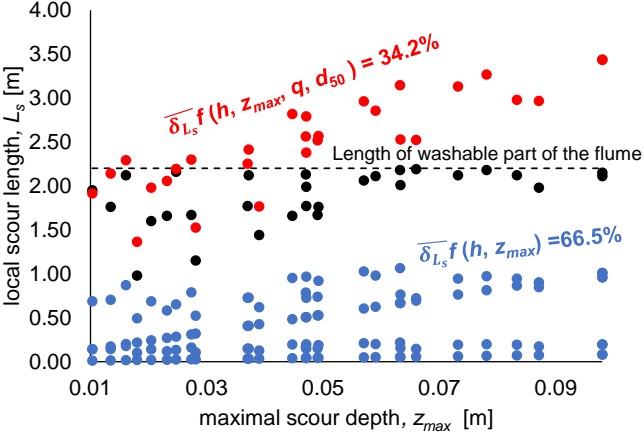

**Figure 10 Local scour length $L_s$ measurements and calculations results.**

Formulas that depend only on the local scour hole depth $z_{max}$ (Shalash and Franke –Eqs. (4) and (5); Müller –Eqs. (7) and (8)) or on the local scour hole depth $z_{max}$ and the depth of water above the unwashed bottom $h$ (Müller –Eqs. (10) and (11)) demonstrated mean relative error of 56.9–72.8% in the scope of total scour length $L_s$ and 38.3–57.0% for the distance between the deepest point of scour and the end of reinforcement $L_e$. The Tajarmovič equation (Eq. (17)) indicates a 392.7% error. Medium relative error for simple formulas (4,7,10) was equal to 66.5% and for formulas (5,8,11 and 17) was equal to 133.3%.

Calculations using the formula, involving not only local scour hole depth $z_{max}$ and the depth of water above the unwashed bottom $h$, but also grain characteristics, represented by $d_{50}$ diameter and hydraulic parameter, i.e., unit water discharge $q$ (the Straube formula –Eqs. (13) and (15)) provide the best fit to the measurement data. The relative error was 34.2% for total scour length and 32.1% for the distance between the deepest point of scour and the end of reinforcement. The Figs. 10 and 11 demonstrate the results of calculations in relation to the measured values.
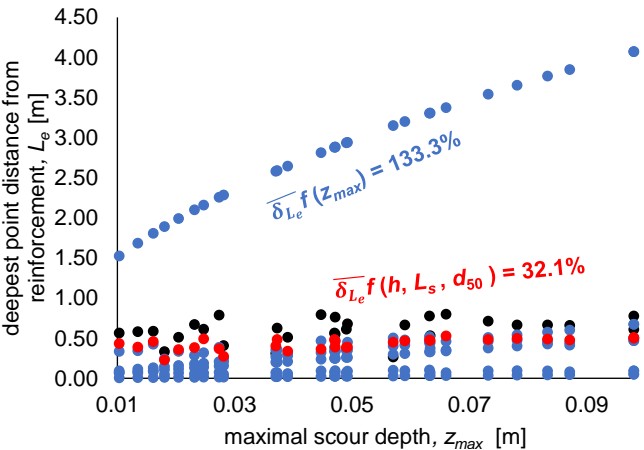

**Figure 11** **The distance between the deepest scour point and the end of reinforcement $L_e$ measurements and calculations results.** (A) Side view; (B) the view from the upper stage of the weir (own elaboration after *Urbański & Hejduk, 2014*).

One combination of parameters *a, b, c, d* and one combination of parameters *k, m, p* were selected basing on the presumed criteria: the formulas described by these parameters were characterized by the lowest average relative error $\delta$ for all 29 test series. The best data explanation for laboratory database of I and II model was achieved in the following parameters values: $a = 7.41; b = 0.38; c = -0.10; d = -0.45; k = 0.34; m = 0.01; p = -0.01$, thence the identified formulas, can be described as:

$$L_s = 7.41 q^{0.38} (z_{max} + h) d_{50}^{-0.10} h^{-0.45} \tag{20}$$

$$L_e = 0.34 L_s d_{50}^{0.01} h^{-0.01} \tag{21}$$

Optimization revealed a diminished error, both in the case of total scour length $L_s$ (10.1%) and for the distance between the deepest point of scour and the end of reinforcement $L_e$ (18.2%).

Verification of the optimized Straube formula was performed on independent data published in 2010 by Gaudio and Marion. In 1998, research was carried out in the Wallingford Ltd., a hydraulic laboratory, on the evolution of local scouring downstream the bed sills cascade. The flume consists of a 60 cm-wide, 24.5 cm-high and 5.57-m working section with rectangular cross-section. For the full description of the duct, see *Gaudio & Marion (2003)*. The bed sills used in all experiments were 25 mm-thick by 15 cm-high wooden plates, with the same width as the transversal section. The sediment used in all tests was sand with median diameter $d_{50} = 1.8$ mm. No sediment recirculating system was adopted.

The similarity of Gaudio and Marion test stand and the flume, where present research was performed, come down to used bed material (sand), the shape of the flume (rectangular, 60-cm width), the same order or magnitude of unit discharges and the transversal type of

**Table 6  Hydraulic flow parameters and local scour properties in *Gaudio & Marion (2003)* model experiments with Straube formula verification.** Where: $q$, unit water flow discharge; $h$, water depth before scour formation; $z_{max}$, maximal scour depth; $L_s$, total scour length; $\delta$, relative error.

| No of test | $q$ | $h$ | $z_{max}$ | $L_s$ | Straube original (Eq. (13)) $L_s$ | Straube optimized (Eq. (20)) $L_s$ | Straube original (Eq. (13)) relative error $\delta$ | Straube optimized (Eq. (20)) relative error $\delta$ |
|---|---|---|---|---|---|---|---|---|
| - | m² s⁻¹ | [m] | [m] | [m] | [m] | [m] | [%] | [%] |
| 1 | 0.020 | 0.050 | 0.084 | 1.25 | 2.10 | 1.63 | 68.4% | 30.1% |
| 2 | 0.032 | 0.065 | 0.097 | 1.65 | 2.71 | 2.09 | 64.5% | 26.6% |
| 3 | 0.021 | 0.050 | 0.071 | 1.35 | 1.93 | 1.50 | 43.3% | 10.8% |
| 4 | 0.025 | 0.062 | 0.068 | 1.47 | 2.03 | 1.56 | 38.1% | 6.1% |
| 5 | 0.021 | 0.050 | 0.058 | 1.43 | 1.73 | 1.34 | 20.7% | 6.6% |
| 6 | 0.028 | 0.071 | 0.083 | 1.25 | 2.37 | 1.81 | 89.9% | 45.1% |
| 7 | 0.030 | 0.070 | 0.095 | 1.90 | 2.62 | 2.01 | 38.0% | 5.7% |
| 8 | 0.021 | 0.050 | 0.087 | 1.50 | 2.19 | 1.69 | 46.0% | 12.9% |
| 9 | 0.024 | 0.060 | 0.102 | 1.55 | 2.53 | 1.94 | 63.0% | 25.3% |
| 10 | 0.030 | 0.070 | 0.110 | 1.63 | 2.86 | 2.19 | 75.5% | 34.4% |
| 11 | 0.027 | 0.065 | 0.084 | 1.98 | 2.35 | 1.80 | 18.6% | 9.0% |
| 12 | 0.020 | 0.060 | 0.069 | 1.52 | 1.88 | 1.44 | 23.9% | 5.1% |
|  |  |  |  |  | Mean relative error |  | 49.2% | 18.1% |

water structure. The main difference is duration of each experimental series: in Gaudio and Marion, experiment series last much longer than the present one: from 45 to 90 h. Gaudio carried out 12 series of measurements in order to obtain the geometric dimensions of local scour holes formed under given hydraulic conditions. The maximal depth $z_{max}$ and the total length of the scour $L_s$ was studied. The hydraulic parameters of each measurement series and the geometric properties of the local scour are summarized in Table 6. Based on the results received, Straube formula was verified in its original and optimized in form by Monte Carlo sampling procedure (Eqs. (13) and (20)). A mean relative error $\delta = 49.2\%$ was obtained for the original form, whereas the application of the optimized Straube formula demonstrated a better description of the data obtained in the laboratory, characterized by an error mean relative $\delta$ equal to 18.1%.

## DISCUSSION

The optimized Straube formula demonstrated very accurate laboratory dataset description, whereas the remaining equations analysis showed a relative error ranging up to more than 390%. The Straube equations forms, which have been optimized for the laboratory workstation, have been validated for field data. The weir on Zagożdżonka River in Czarna (Poland) was built in the fifties of last century as a concrete hydraulic structure to store up the water and to use its energy to drive the mill wheel. The total width of the spill is divided by I-beam guides into 3 clear spans: 1.16 m wide outermost spans and 1.22 m center span (Figs. 12A, 12B). In the guides, a measuring sharp-crested triangular weir was installed. The height and shape of the weir edges were developed to ensure non-submergence weir working conditions at the highest possible flow rates.

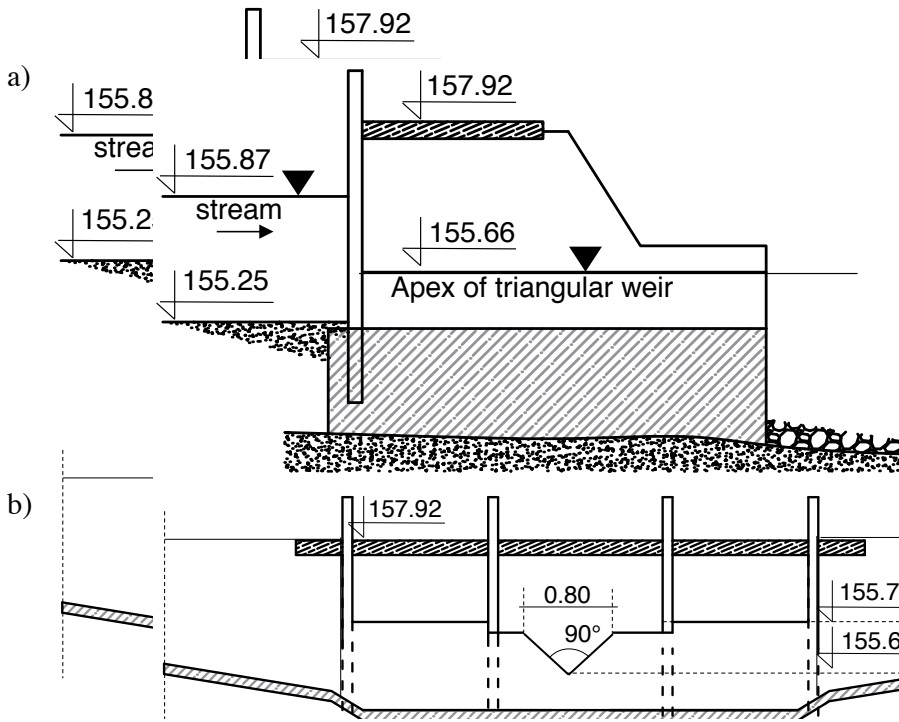

**Figure 12 Weir in Czarna schematic.** (A) Side view; (B) the view from the upper stage of the weir (own elaboration after *Urbański & Hejduk, 2014*).

The downstream part of the structure consists of 8.80 m long concrete reinforcement with a longitudinal slope of 1%, followed by a 0.60 m drop, so it could be recognized as similar to laboratory condition test models. The river bed, directly below the drop, is partially covered with a stone riprap over on a reach of about 1.0 m, and in a further section it is scourable, made of sand, with $d_{50}$ diameter of 0.42 mm and $d_{90}$ diameter of 0.74 mm.

On June 11 2013, a flood occurred. The flow rate in the hydrograph peak reached 5.06 m³s⁻¹. This event resulted in local scour formation downstream the weir, whose dimensions were measured, analyzed and published by *Urbański & Hejduk (2014)*. Field measurements performed the following local scour dimensions:

–water depth above the deepest scour point $H_{max} = z_{max} + h = 2.43$ m;

–local scour length $L_s = 13.8$ m;

–the distance between the deepest point of scour and the end of reinforcement $L_e = 5.20$ m.

In the case of water depth $H_{max}$ calculations an error of 39.5% was achieved using the Rossinski formula with a $k_1$ parameter equal to 1.70 (Eq. 1) (Table 7). The best fit of the measurement and calculations was obtained for Müller equations, where measured scour length and the distance between the deepest point of scour from the end of reinforcement rare within the ranges described in equations no. 8 and 10 (an error of 0%). In the case of default form of Straube equations, 57.2% of an error was achieved for scour length $L_s$ analysis and 7.7% for the $L_e$ distance.

**Table 7  Field measurements and calculations results summary table—Czarna Gauge.** $z_{max}$,maximal scour depth; h, water depth before scour formation; $L_s$, scour length; $L_e$, the distance between the deepest point of the hole and the end of reinforcement.

| | Geometric scour parameters | | |
| --- | --- | --- | --- |
| | $H_{max} = z_{max} + h$ [m] | $L_s$ [m] | $L_e$ [m] |
| Field measurements | 2.43 | 13.8 | 5.2 |
| Author: | Calculations results using Eqs. (1), (7)–(21) (error %) | | |
| Rossinski | (1) 3.39 (39.5%) | | |
| Müller | | (7) 10.3–12.1 (12.3%) | **(8) 5.0–6.1 (0%)** |
| Müller | | **(10) 11.6 –17.5 (0%)** | (11) 5.7–8.6 (9.6%) |
| Straube | | (13) 21.7 (57.2%) | (15) 4.8 (7.7%) |
| (Optimized) Straube | | **(20) 13.8 (0.2%)** | (21) 4.3 (16.6%) |
| Tajarmovič | | | (17) 4.8 (159.6%) |

An optimized form of Straube equations (20) and (21) were checked on the field measurements. Calculations using the Straube's optimized formula showed excellent adherence for the measured and calculated value of the local bottom scour length (an error equal to 0.2%). However, the distance of the maximum hole depth from the end of the reinforcement was underestimated and the underestimation amounted to 16.6% of this value. A common observation for laboratory and field tests is the overestimation of both parameters using the Tajarmovič formula.

## CONCLUSIONS

Two gated check models were investigated in water discharge flowing out from under the gate, characterized by different roughness of the reinforcement downstream, followed by scourable bed. 29 measurement series were performed in total, each lasting 8 h. The basic geometrical parameters of local scour hole, resulting from the disturbance of hydrodynamic balance of the system were examined using autonomic remote-controlled measuring unit. The construction of the tested models was chosen due to the prevalence of such solutions among real objects.

10 computational formulas, used for many years in the water engineering practice, were verified for laboratory data. It was stated that functions based only on one ($z_{max}$) or two ($z_{max}$, h) parameters provide weaker adjustment between calculations results and laboratory measurements. The Straube's formula, assuming that geometric parameters follow up on not only maximal scour depth and water level, but also granulometric parameters, represented by medium grain diameter $d_{50}$ and hydraulic properties of experiment, such as unit discharge q was distinguished as the best description of laboratory test results.

The Straube function demonstrated the mean relative error of 34.2% in the case of comparing the measurement and calculation result of the local scour depth and an error

of 32.1% for the distance of the deepest point from the end of the reinforcement, while medium error for all the rest of formulas was 67% for $L_s$ and 133% for $L_e$.

The Monte Carlo sampling method allowed the original formulas to be adapted to calculate the local scour geometrical parameters downstream the model of the gated check. In optimizing the parameters, the criterion of minimizing the relative error was applied. The Monte Carlo sampling procedure resulted in a much better match between the calculation results and the dimensions measured in the laboratory: Straube function optimized in this way demonstrated an error of 10.1% in the case of comparing the measurement and the calculation of the local scour length and an error of 18.2% for the distance of the deepest point from the end of the reinforcement.

The Straube's formula, chosen as the best describing laboratory results, was verified on independent dataset, whose main features in common with present experiment characteristics are: used bed material (sand), the shape of the flume, the same order or magnitude of unit discharges and the transversal type of water structure. A mean relative error $\delta = 49.2\%$ was obtained for the original form of the Straube formula, and 18.1% for optimized formula using the Monte Carlo sampling method. Due to the data availability, only the total length of the scour was compared.

The optimized for laboratory measurements equation was checked for the real object, which was selected on the basis of the similarity of the downstream reinforcement, and of the data availability. It should be emphasized that field measurements of the bottom shape after the formation of local scour hole are often difficult to access due to the imperfection of measuring instruments and lack of data before the formation of a local scour. The optimization led to obtain an error of 0.2% for scour length and an error of 16.6% for the distance of the deepest point from the end of the reinforcement.

The extension of the optimized Straube formula verification to other hydro-technical field objects is necessary for the applicability of the investigation, however it has to be stated that very high degree of adjustment of calculation results to field data (especially local scour length) provide an encouraging premise for further research.

### Funding
This work was supported by the Department of Hydrotechnics and Technology and the Department of Water Engineering and Applied Geology of Warsaw University of Life Sciences. The funders had no role in study design, data collection and analysis, decision to publish, or preparation of the manuscript.

### Grant Disclosures
The following grant information was disclosed by the authors:
Department of Hydrotechnics and Technology.
Department of Water Engineering and Applied Geology of Warsaw University of Life Sciences.

## Competing Interests

The authors declare there are no competing interests.

## Author Contributions

- Marta Kiraga conceived and designed the experiments, performed the experiments, analyzed the data, prepared figures and/or tables, authored or reviewed drafts of the paper, and approved the final draft.
- Zbigniew Popek conceived and designed the experiments, authored or reviewed drafts of the paper, and approved the final draft.

## Data Availability

The raw data is available as a Supplemental File.

## Supplemental Information

Supplemental information for this article can be found online at http://dx.doi.org/10.7717/peerj.10282#supplemental-information.

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
