# Peer review of "On local scouring downstream small water structures"

_PeerJ, doi:10.7717/peerj.10282_

## Round 0.1 · original submission · Major Revisions

We have received three significant reviews, all of which recommend major revision. In my opinion, although there are many comments that require attention, none should be problematic to the authors. Please provide detailed responses to each reviewer comment, indicating how the comment was addressed in the revised manuscript. One theme to pay particular attention to is the generality of the results; multiple reviewers raised questions along these lines and the authors should pay attention to properly framing their conclusions accordingly.

·

Basic reporting

The article must be written in English and must use clear, unambiguous, technically correct text. The article must conform to professional standards of courtesy and expression.
After reading the article, in my opinion the English language in the text is far from perfect. Although English is not my native language, I have the impression that the sentence structure and some expressions used in the article correspond to the structure of the Polish language. Undoubtedly, the text requires proofreading and the naming of technical terms by a native speaker. The parts of text and phrases that I have doubts about are marked in yellow in the text (PDF).

The article should include sufficient introduction and background to demonstrate how the work fits into the broader field of knowledge. Relevant prior literature should be appropriately referenced.
The subject matter of the publication concerns the technical aspects of the operation of hydro-engineering structures. It discussed the impact of a small weir on the stability of the river bed downstream the water structure and the formation of local scour. The article was submitted to the Environmental Science section of PeerJ Journal. The authors noted that the content of the article falls within the scope of the following topics: Coupled Natural and Human Systems, Natural Resource Management, Ecohydrology, Environmental Impacts Food, Water and Energy Nexus. In my opinion, the introduction lacks an analysis of the relationship between the main topic of the article (local scouring) and the listed topics of the Environmental Science section. It would be necessary to do additional research showing the relationships between them. Research in world literature should be significantly deepened in this area.

The structure of the submitted article should conform to an acceptable format of ‘standard sections’. Significant departures in structure should be made only if they significantly improve clarity or conform to a discipline-specific custom.
The article structure is correct.

Figures should be relevant to the content of the article, of sufficient resolution, and appropriately described and labeled.
The quality of the drawings is acceptable. The charts 9 and 10 are too small. Horizontal axes should be definitely longer than vertical. In addition, the information on these charts is not clear. For example, it is not known what the fields colored blue and red are. It is not known how to interpret the black horizontal dotted line (bottom edge, Fig. 9). In my opinion the figures 9 and 10 should be improved.

Experimental design

The submission must describe original primary research within the Aims & Scope of the Journal.
The article describes the original results of laboratory tests in the field of water engineering. The authors did not explicitly indicate the relationship of their research with the domain of Environmental Sciences. I leave the decision to the Editor whether the topic of the work falls within the journal area of interest (Aims & Scope). In my opinion it would be needed to do additional research showing the relationships of authors’ research with the field of Environmental Sciences before acceptance.

The submission should clearly define the research question, which must be relevant and meaningful. The knowledge gap being investigated should be identified, and statements should be made as to how the study contributes to filling that gap.
The research question is clearly defined. The paper focuses on the verification of some empirical formulas to estimate the scour dimensions in the case of local scouring processes downstream the small hydro-engineering structure (weir). The knowledge gap being investigated was identified as a uncertainty of the most often used formulas.

The investigation must have been conducted rigorously and to a high technical standard.
Laboratory experiments were carried out using well known measurement methods with appropriate measurement procedures. High-class measuring equipment with sufficiently good accuracy was used.

Methods should be described with sufficient information to be reproducible by another investigator.
The test stand and the course of the experiments were described in detail. It is possible to repeat it by other researchers.

Validity of the findings

The data should be robust, statistically sound, and controlled.
The article presents the results of 29 measurement series. The way experiments are carried out and their results are beyond doubt. However, according to the research description, each experiment was carried out only once. This means that the authors have not examined (or do not mention in this article) the issue of repeatability of the results obtained. If the experiments were not repeated, then there is doubt as to their statistical quality. This requires explanation and commentary from the authors.

The data on which the conclusions are based must be provided or made available in an acceptable discipline-specific repository.
The authors have included all measurement data in a shared database. However, in some places there are descriptions in Polish, which makes data analysis difficult. All descriptions should be in English.

The conclusions should be appropriately stated, should be connected to the original question investigated, and should be limited to those supported by the results. In particular, claims of a causative relationship should be supported by a well-controlled experimental intervention. Correlation is not causation.
There is some doubt about the conclusion about the optimized Straube formula. All available (made by the authors) measurement data were used to determine (optimize) the parameters of this formula. This process can be called identification. Next, validation of the optimized formula was performed based on field test (Czarna Weir). Unfortunately, the research process lacked an intermediate step - verification of the obtained formula. Such verification should be carried out before validation, using independent measurement results other than those used for the identification. Authors should fill this gap in the research procedure.

Additional comments

Some remarks
As a result of optimization (identification) of the Straube formula, the authors received positive values of parameters c, d and p (equations 18, 19 and 20, 21). According to the original formula (equations 13.15), their value is negative. This seems a mistake.
All units should be written using the exponential form. There is a lot of wrong forms in the paper.
Authors should avoid citing Polish literature if it does not concern publication of research carried out by Polish researchers. It is not allowed to cite and place in the bibliography publications with titles in Polish - Dabkowski et al., 1982. For classic formulas, the first sources should be given.

·

Basic reporting

The overall reporting within this paper appears sound. There are some clarifications can be made throughout the text to improve clarity and understandability of the work.

The abstract and body of the paper both reference “many years” of disagreement on parameters that influence size and depth of scouring but does not provide sufficient evidence of this ongoing disagreement (whether through published papers or through reporting other evidence). A discussion of why these previous papers could not come to the same conclusions as this paper would be helpful

Please also see comments to the authors.

Experimental design

The experimental design of the paper is clear, although would benefit from being introduced earlier in the text and the abstract and introduction referencing the intended work more specifically. Given the several components (laboratory work, field work, and several stages of analysis, a clear description of the procedures from beginning to end would make the work more easily replicated.

Please also see comments to the authors

Validity of the findings

The paper makes many good points and has a good experimental design with the overall framework supported by the material presented. However, there are some items presented that could benefit from additional clarification, to help prevent unintentionally misrepresenting the paper as more than it is. As discussed in “comments to the author,” the field evaluation of these equations would be greatly improved by either adding evaluation of additional structures and events, or by caveating the conclusions that these conclusions were tested against a single data point.

Additional comments

- Overall, my opinion is that this is a generally well-written paper with the potential to add to the literature. With revision, I believe this paper will be suitable for publication in this journal.
- Line 68: This first sentence of the introduction is confusing as written. It refers to “the river” several times but does not indicate which river (to speak generally, likely the correct phrasing would be by discussing “rivers” broadly). There also appears to either be a word missing or some other issues, as “structures unavailable influences” is likely supposed to be either “structures causes unavoidable influences on…” or perhaps “structures unavoidably influences”.
- Lines 69-71: These lines discuss several impacts of these structures, and are written as absolutes (that is to say, that these impacts *always* occur). It is not clear if that was intended or if these are meant to be common or likely (but not certain) impacts. Additionally, this list of impacts is not common knowledge and should be properly cited.
- Lines 71-72: Especially at its first use, it may be preferable to clarify that “directly below” the structure refers to the area just downstream, as opposed to the area underneath the structure.
- Line 73: Is “permanently” supposed to be “potentially” or some other word here? It would appear that although this damage can be considerable, isn’t necessarily permanent.
- Line 89: Please define “recognition laws” and if appropriate provide a reference.
- Line 90-92: This section refers to seeking “universal principles” but does not acknowledge that the variety of hydraulic and built-environment conditions means that it’s possible that universal parameters may not be appropriate to this concern. These lines also discuss “many experiments” related to these concerns but does not cite said experiments
- Line 98: The use of “on the other hand” should have a phrase before it with “on one hand”. Otherwise, consider different phrasing.
- Line 117: This sentence discusses “chosen empirical formulas” but does not state how said formulas were chosen.
- Line 125: The phrase “basically comes down to” is a very casual phrase. Consider replacing with a more formal description of measurements made.
- Lines 128-131: This section describes a series of parameters for the structure of the experimental model water structure. Although these descriptions are welcome, they do not explain how the various parameters were chosen. The paper would be made stronger by providing these reasonings.
- Lines 168-171: This section provides a good description of LiDAR technology, but does not cite any references. I recommend referencing relevant work. Additionally, I believe that with appropriate references to information about LiDAR, Figure 6 will no longer be necessary and can be removed.
- Lines 176-177: Although the description of the sustainable properties of the polyesters used is intellectually interesting, I’m not sure this description is relevant to the research and may be more distracting than helpful.
- Lines 213-217: This section lays out a series of formulas. Although various formulas were noted as being from different authors (e.g. Shalash and Franke) a single source is cited for all equations. I recommend citing the original source for each of these groups of equations.
- Lines 292-310: It appears that this paper’s laboratory findings were validated against a single high flow event at a single water structure to demonstrate the correlation between modeled and measured scouring length and depth. Because the laboratory data is only compared to a single field example, the authors should consider reviewing all of their conclusions to assure they are not portrayed as being broadly applicable. Before the optimized equations could be considered to be representative, they would need to be compared to many events at many facilities at many events.

Reviewer 3 ·

Basic reporting

Basically, this study presents an optimization of current gate scour prediction formula coupling with the Monte Carlo method. The presentation is proper and clearly understood. Besides, the authors themselves developed a laser scanning to detect the shape of the scoured hole downstream the gate, which is interesting.

Experimental design

no comment

Validity of the findings

no comment

Additional comments

I think that the authors need to elaborate how the install the laser scanning technique. I personally recommend the publication as long as the authors can commit a major revision with respect to the following comments.
- Please cite the recent-5-year studies in the introduction to show the progresses of the local scour due to a gated check.
- Line 44-47: Please rephrase the statement. It is not clearly stated.
- Line 48: Please rephrase “researchers still disagree on the parameters that influence its size and intensity”.
- Line 49: the authors say that there are no universal methods for estimating its magnitude. Does this mean that universal methods have been developed in this study?
- Line 51: what is the “great dams”?
- Line 58: “was applied to describe the shape”. “describe” is not proper here. Please revise.
- Line 63-64: “basing on Straube formula” should be “basing on the Straube formula”. Also, please pay attention to any place of the same issue.
- Line 68: “unavoidable influences” should be “unavoidably influences”.
- Line 70: “Simultaneously erosion” should be “Simultaneously, erosion”. Similarly, please check the similar issues through the entire context.
- Line 74: “The increased erosion of a riverbed is an unfavorable and undesirable phenomenon not only due to the slow degradation of the riverbed”. Not only … but also… Please rephrase.
- Line 124-125: Why the flume is with no bed inclination downstream? This actually is not true for the real case. Please clarify the intention of this design consideration.
- Line 129-131: What’s the consideration of “the bottom was non-washable”? Is there any existing study for reference?
- The flume configuration: the sandy bed which the authors used the water to scour has a limiting downstream length. However, for a real case, the length should be long enough for scour. The downstream should have no impact on the scour pattern. So what’s the authors’ consideration?
- What’s the drainage function with a pipe for?
- Line 135: “A pin water gauges were used” Grammar error. The authors need do proof-reading for the entire manuscript.
- Line 137: what’s the function of the moving disc probe besides the laser scanner device
- Line 137: “sandy bottom was measured with laser scanner device” should be “sandy bottom was measured with a laser scanner device”. Also revise the similar issues for the entire context.
- Line 139: the unit of roughness coefficient is problematic.
- Line 140: “during the studies” should be “during the study”.
- Line 142: “the following range Qw = 0,010 – 0,045” should be “0.010 - 0.045”.
- Line 150: should be “due to the flow resistance”. Again, please pay attention to the use of “a, an, the” for the entire manuscript.
- Line 188: zmax should be the elevation not the depth. Confused.
- Line 195: should be “finer than the reported particle size” not “under the reported particle size”.
- Line 214: “ ” this way is not scientific for a research paper.
- Line 220-227: the description of the Monte Carlo method to evaluate the optimal parameters of the prediction formula is poorly stated. The authors need to detail how the authors used the Monte Carlo method for their intention, for instance the procedures.

---

## Round 0.2 · accepted · Accept

Thank you for your careful attention to the reviewers' suggestions in your revised manuscript.

·

Basic reporting

no remarks

Experimental design

no remarks

Validity of the findings

no remarks

Additional comments

Dear Authors and Editors,

I read the letter of the authors of the article with great pleasure. I am pleased to say that the authors took into account all my suggestions for changing the structure of the article and supplementing it with a part related to the verification of their calculations on independent material. In addition, the section on indicating the relationship between the research and the field of environmental engineering has been significantly developed. There is also a visible improvement in the quality of technical English in the field of hydraulic engineering.

Finally, I must say that I am very impressed with the work the authors had to put into improving their article. I hereby recommend the revised article for publication in the PeerJ Journal.

In the attachment I am sending a pdf file with two marked places where I noticed minor errors.

Regards
Michał Szydłowski